# Young Adults with a Parent with Dementia Show Early Abnormalities in Brain Activity and Brain Volume in the Hippocampus: A Matched Case-Control Study

**DOI:** 10.3390/brainsci12040496

**Published:** 2022-04-13

**Authors:** Ian M. McDonough, Christopher Mayhugh, Mary Katherine Moore, Mikenzi B. Brasfield, Sarah K. Letang, Christopher R. Madan, Rebecca S. Allen

**Affiliations:** 1Department of Psychology, The University of Alabama, 505 Hackberry Lane, P.O. Box 870348, Tuscaloosa, AL 35487, USA; cmayhugh@crimson.ua.edu (C.M.); mkmoore6@crimson.ua.edu (M.K.M.); mbbrasfield@crimson.ua.edu (M.B.B.); sletang@crimson.ua.edu (S.K.L.); rsallen@ua.edu (R.S.A.); 2School of Psychology, University of Nottingham, Nottingham NG7 2RD, UK; christopher.madan@nottingham.ac.uk

**Keywords:** connectome, dementia, hippocampus, magnetic resonance imaging, resting state

## Abstract

Having a parent with Alzheimer’s disease (AD) and related dementias confers a risk for developing these types of neurocognitive disorders in old age, but the mechanisms underlying this risk are understudied. Although the hippocampus is often one of the earliest brain regions to undergo change in the AD process, we do not know how early in the lifespan such changes might occur or whether they differ early in the lifespan as a function of family history of AD. Using a rare sample, young adults with a parent with late-onset dementia, we investigated whether brain abnormalities could already be detected compared with a matched sample. Moreover, we employed simple yet novel techniques to characterize resting brain activity (mean and standard deviation) and brain volume in the hippocampus. Young adults with a parent with dementia showed greater resting mean activity and smaller volumes in the left hippocampus compared to young adults without a parent with dementia. Having a parent with AD or a related dementia was associated with early aberrations in brain function and structure. This early hippocampal dysfunction may be due to aberrant neural firing, which may increase the risk for a diagnosis of dementia in old age.

## 1. Introduction

Alzheimer’s disease and related dementias (ADRD) are a public health challenge due to the incurable nature of these chronic neurocognitive disorders [1,2]. Early stages of Alzheimer’s disease (AD) are characterized by the aggregation of amyloid-β (Aβ) plaques and the formation of neurofibrillary tangles in the brain [3,4]. Accumulating evidence suggests that AD begins with neuropathological changes decades before noticeable symptoms begin [5,6]. One way to better understand the AD process is by investigating intergenerational factors that confer risk of the disease upon subsequent generations. The present study takes a lifespan perspective by assessing brain markers in offspring of parents who have ADRD.

The majority of research bridging links with early progression of AD and inherited factors have focused on the apolipoprotein E (APOE) gene. People with at least one APOE ε4 allele sometimes have faster aggregation of Aβ [7,8], more neurodegeneration of brain structures [9,10], and can have an earlier onset of the disease [11,12]. These findings are consistent with the idea that among older adults, having the APOE ε4 allele increases the progression of the disease. Only a few studies have investigated whether this genetic risk occurs earlier in the lifespan such as in young adulthood. At these younger ages, AD pathology has not yet accumulated [13] and evidence for differences in brain structure due to gene status is mixed [14,15,16]. However, researchers have been able to find early effects of genetic status on brain function. A recent review investigating differences in resting-state connectivity in young and middle-aged adults found some evidence for increases in functional connectivity in adults at risk for AD (primarily APOE ε4 carriers), especially in the default mode network (DMN) [17]. However, the authors also noted that little consensus was found on the directionality of the effects. Similarly, a recent review investigating risk factors for AD in task-evoked functional magnetic resonance imaging (fMRI) in older adults found greater activation for APOE ε4 carriers than non-carriers also in the DMN mostly in the left hemisphere [18]. Importantly, these genetic differences were also found for mean activation of the hippocampus during a memory task using fMRI in young adults [15]. Together, these results implicate a clear genetic risk factor that leads to aberrant functional activity and sometimes smaller brain size as early as young age.

While the APOE ε4 allele appears to be a potential forecaster of AD [19,20,21], APOE ε4 has been connected to less than 40% of AD cases, suggesting that other reasons, such as having a family history of AD, might be predictors of the disease on their own [17,22,23]. To date, having a family history of AD has not been studied as much as the risk for having the APOE ε4 allele. In fact, research investigating both factors has sometimes shown that impact of a first-degree relative of AD impacts brain structure and brain function to a larger extent than APOE status alone. These family history effects have been found in middle-aged and older adults for cortical thickness in the medial temporal lobes [9] and gray matter volume in frontal, parietal, and temporal cortices [24]. In regard to brain function, Bassett et al. [25] found that middle-aged and older adults (aged 50–75) with a parent with AD exhibited increases in task-related brain activity using fMRI in temporal regions including the hippocampus when controlling for APOE status. More recently, functional connectivity has also been found to be reduced in the medial temporal lobes (MTL) in older adults with a family history of AD after controlling for APOE status [26].

These studies point to the possibility that brain activity serves as the earliest marker of abnormal function and often originates initially in the medial temporal lobe such as the hippocampus [27]. The hippocampus appears to be selectively vulnerable to AD risk factors and early AD pathology [18,27,28,29]. The hippocampus is also highly connected to the DMN [30,31,32], where early accumulation of AD pathology and neurodegeneration occur [28]. However, whether brain activity evidences an increase or decrease and when in the course of the disease these alterations occur are less clear [17,18]. Sperling and her colleagues [3,29] have observed that early stages of the disease process (i.e., preclinical and early mild cognitive impairment [MCI]) are often characterized by increases in MTL activity, but are characterized by decreases in MTL activity during later stages of the disease process (i.e., late MCI and AD). Supporting this pattern, greater hippocampal activity at baseline in older adults has predicted steeper longitudinal declines in cognition [33] and greater future decline in hippocampal activity, suggesting an inverted “U” pattern of brain activity in the MTL. A recent review of fMRI studies in older adults suggested that this pattern might depend on the memory stage being assessed (encoding or retrieval) [18].

The reason for early aberrations in MTL regions is still unknown. One proposal is that since much of the hyperactivity is found in brain activity while undergoing a memory task, the increase is compensatory [34]. That is, individuals at risk for AD might need to recruit regions in the MTL to shore up additional neural resources, which may be needed when other brain regions are starting to show structural decline. Similar arguments have been made for increases in brain activity found as a function of normal aging [35]. However, an alternative proposal is that this aberrant activity is a sign of early synaptic dysfunction and/or inefficiency [36] and might actually be toxic [37,38]. Supporting this latter idea, increases in AD pathology (e.g., Aβ and tau) have also been linked to aberrant hyperactivity [39].

Note that while a couple of studies have investigated early-lifespan impacts of APOE on brain function and structure, studies investigating a family history of AD have been limited to middle and older ages (ages ≥ 38) [17]. Given that previous research has suggested that early stages of AD are characterized by hyperactivity in the hippocampus, we predicted that this pattern could manifest itself in the resting blood-oxygen-level-dependent (BOLD) signal. Wig and colleagues [40] have shown how simple measurements of mean resting BOLD activity in the MTL can be sensitive to individual differences in episodic memory performance in young adults. However, new research has also suggested that variability of the BOLD signal could be a novel marker of brain health [41]. One common measure of variability is the standard deviation of the BOLD signal [42,43], where a larger standard deviation of the BOLD signal is associated with better health and cognition [41]. Lastly, we tested whether estimates of hippocampal volume might also be different between the groups with and without a family history of AD, adding evidence that these differences are widespread.

We tested these hypotheses in a group of young adults aged 22 to 35 who have a parent with late-onset ADRD compared with young adults who do not. Given that most people are middle-aged by the time their parent receive a diagnosis of ADRD, this sample provides a rare glimpse as to how early such brain aberrations might occur. To the extent that having a parent with ADRD passes on a genetic risk or encourages an environment or lifestyle that enhances one’s risk of ADRD, subtle signs of unusual hippocampal brain activity might already be apparent. 

## 2. Materials and Methods

### 2.1. Participants

Participants from the Human Connectome Project (HCP) were used. The goals of the HCP are to explore human brain circuits [44]. Participants were young adults that were relatively healthy and free of a prior history of significant psychiatric or neurological illnesses, but could have a history of smoking, heavy drinking, or recreational drug use. All participants gave their written informed consent for inclusion before they participated in this study. This study was conducted in accordance with the Declaration of Helsinki, and the protocol was approved by the Ethics Committee of Washington University, St. Louis and The University of Alabama (IRB#16-OR-292, initially approved on 25 October 2016).

Out of 1094 participants in the full sample with resting-state fMRI data, 14 of the participants self-reported having at least one parent with ADRD (1 with both parents, 3 with mother only, and 10 with father only). Note that the question posed to participants was not exclusive to AD because neurologists often do not need to determine a specific diagnosis to proceed with recommendations, especially given that there is no cure and only short-term pharmaceutical solutions. We focus on the most-studied brain regions, which are, in the context of AD, the most common. This group was matched to participants without a parent with ADRD (see Statistical Analysis section). Sample characteristics can be found in Table 1.

### 2.2. Measures of Cognition

While the HCP data include multiple cognitive measures [45], the present study only reported three measures that were most germane to this study: verbal episodic memory, visual episodic memory, and reading ability. The Penn Word Memory Test [46] measured verbal episodic memory by presenting participants with 20 words followed by an old/new recognition task with 20 additional distractors. The dependent variable included total number of correct responses. The Picture Sequence test [47] measured visual episodic memory by presenting participants with 15 pictures of objects and activities that were thematically related followed by a test that had participants order the pictures in their correct sequence. The dependent variable is derived from the cumulative number of adjacent pairs of pictures remembered correctly over 3 learning trials. The Oral Reading Recognition Test [48] measured reading ability by having participants pronounce low-frequency words with irregular orthography. The dependent variable included the total number of correct responses. 

### 2.3. MRI Acquisition and Preprocessing

All data were acquired on a Siemens Skyra 3T scanner housed at Washington University in St. Louis. The scanner had a customized SC72 gradient insert and a customized body transmitter coil with 56 cm bore size (diffusion: Gmax = 100 mT/m, max slew rate = 91 mT/m/ms; readout/imaging: Gmax = 42 mT/m, max slew rate = 200 mT/m/ms). The HCP Skyra had the standard set of Siemen’s shim coils (up to 2nd order) and used Siemen’s standard 32-channel head coil. BOLD fMRI data were acquired using a T2*-weighted gradient-echo EPI sequence with 72 axial slices per volume, 104 × 90 matrix (2.0 × 2.0 × 2.0 mm^3^), FOV = 208 mm, TE = 33.1 ms, and TR = 720 ms, FA = 52°. Across four scanning sessions of 15 min each, a total of 4800 frames were acquired. Participants were instructed to keep their eyes open and focused on a bright cross hair on a dark background. Across sessions, oblique axial acquisitions alternated between phase encoding in a right-to-left direction (two runs) and phase encoding in a left-to-right direction (two runs).

For the brain structure analyses, postprocessed MRI data were used based on FreeSurfer 5.3.0-HCPFreesurfer [44,49]. This pipeline segmented the volume into predefined structures, reconstructed white and pial cortical surfaces, and performed FreeSurfer’s standard folding-based surface registration to their surface atlas (fsaverage). To assess hippocampal volume, a hippocampal occupancy score was created [50] by taking the values for the left and right hippocampal volumes and dividing them by the sum of the hippocampal volume and the inferior lateral ventricle volume for each hemisphere, separately. The left and right hippocampal occupancy scores were then corrected using intracranial volume estimates from Freesurfer. The hippocampal occupancy score has advantages over simple measures of hippocampal volume because cross-sectional measures of hippocampus volume confound baseline levels (i.e., individual difference) with longitudinal change [50]. 

For the brain function analyses, postprocessed fMRI datasets were used, which consisted of standard processing methods using FSL [51]. Below briefly summarizes the HCP processing pipeline [49]. First, gradient-nonlinearity-induced distortion was corrected for all images. Next, FMRIB’s Linear Image Registration Tool (FLIRT) was used for motion correction using the single-band reference (SBRef) image as the target. The FSL toolbox “topup” [52] was used to estimate the distortion field in the functional images. The SBRef image was used for EPI distortion correction and is registered to the T1w image. One-step spline resampling from the original EPI frames to MNI space was applied to all transforms. Lastly, image intensity was normalized to mean of 10,000 and bias field was removed. Data were cleaned using ICA+FIX [53,54], which included linear detrending, regression of 24 motion parameters, and ICA noise components removed. This method better removes artifacts than regressing out white matter and/or CSF signal directly, as well as using the “scrubbing” method [55]. Global signal was not removed. 

Left and right hippocampi were used as regions of interest (ROIs) because previous research has suggested that these regions are some of the first to show differences in brain activity in the preclinical AD stage [56]. AFNI [57] was used to create the ROIs from the TT Daemon atlas and ROI placement was adjusted using MRIcron (Figure 1a). AFNI was again used to extract the mean and standard deviation (after removing the mean) of the BOLD signal from each ROI in each hemisphere and each of four runs. Data from the four runs were averaged together.

### 2.4. Statistical Analyses

Propensity score matching [58] was used to match the two groups of participants (young adults with a parent with ADRD and those without). This procedure reduces confounds between the comparison groups of interest and is especially suitable for matching uneven sample sizes [59]. First, the entire sample of participants was reduced to only those participants who self-reported having a parent with ADRD. The remaining participants were then matched to this group by creating propensity scores using the MatchIt package [60] in R. Propensity scores were calculated using a logistic regression using Parent with ADRD as the dependent variable and the matching factors of interest as the independent variables: age, years of education, race (non-White, White), ethnicity (non-Hispanic, Hispanic), employment status (not working, part-time employment, full-time employment), number of parents with depression, number of parents with bipolar disorder, sex, and reading ability. The propensity score for each individual was calculated using the person’s predicted probability of having a parent with ADRD, given the estimates from the logistic regression model. Then, pairs of observations that had similar propensity scores, but differed in parental history of ADRD, were matched using the nearest neighbor method, which matched the closest control for each treated unit one at a time [61]. 

Independent *t*-tests were conducted on demographic factors to ensure appropriate matching. To determine how brain activity differed between groups, independent *t*-tests were conducted on (1) the mean BOLD signal and (2) the standard deviation of the BOLD signal for each hemisphere. To determine how brain volume differed between groups, the same analyses also were conducted on the hippocampal occupancy scores. Analyses were also conducted to test whether verbal and visual episodic memory performance differed between groups, also using independent *t*-tests. We used *t*-tests rather than analyses of variance due to the small sample size and our a priori predictions.

## 3. Results

### 3.1. Sample Characteristics

Independent *t*-tests were conducted on the sample characteristics to identify any group differences (see Table 1). No group differences were found (*p* > 0.26).

### 3.2. Hippocamal Brain Activity

As shown in Figure 1b, the mean BOLD signal was greater in the hippocampus for young adults who had a parent with ADRD compared with those without a parent with ADRD. Supporting these observations, we found a significant group difference in the mean BOLD signal for the left hippocampus (t(26) = 2.27, SEM = 175.98, *p* = 0.032, 95% CI = [38.20, 761.66]). We found no significant difference between groups for the right hippocampus, although the direction of the pattern was the same (t(26) = 1.33, SEM = 213.07, *p* = 0.20, 95% CI = [−155.19, 720.76]). The standard deviation of the BOLD signal across the regions was numerically reduced in young adults with a parent with AD compared with those without a parent with AD (Figure 1c). However, none of these group differences were significant (*p* > 0.28).

### 3.3. Hippocampal Volume

As shown in Figure 2a, the left hippocampal occupancy score was smaller in young adults with a parent with ADRD compared to those without a parent with ADRD (t(26) = 2.13, SEM = 0.011, *p* = 0.042, 95% CI = [0.001, 0.045]). No significant difference was found for the right hippocampal occupancy score (t(26) = 1.76, SEM = 0.009, *p* = 0.091, 95% CI = [−0.034, 0.003]). Given these laterality differences, a laterality index was created [(left − right)/(left + right)] and the group differences were highly significant using this index score (t(26) = 3.49, SEM = 0.006, *p* = 0.002, 95% CI = [0.009, 0.033]).

## 4. Discussion

In the present study, we investigated possible early markers of brain abnormalities in young adults aged 22–35 who had a parent with ADRD. We found two key differences in those with and without a parent with ADRD. First, we found that resting brain activity was higher in young adults with a parent with ADRD than young adults without a parent with ADRD. Second, we found that hippocampal volume, as estimated by hippocampal occupancy scores, was smaller in young adults with a parent with ADRD than those without. These findings are the youngest reported differences in the literature between those with and without a family history of ADRD and suggest that having a family member with ADRD changes the neurobiology of one’s offspring much earlier than previously known, thus putting them at risk for later development of the disease. This new knowledge critically impacts the age at which lifestyle interventions might be implemented to alter the course of potential subsequent cognitive decline.

Because of the small sample size of each group, we chose to take a region of interest approach to test this idea in the most well-known brain region that declines in AD: the hippocampus [56,62]. The hippocampus also serves as a key site for the formation and retrieval of episodic memory [63]. We found converging evidence for the role of early hippocampal aberrations in the form of increased resting state activity (i.e., hyperactivity). Importantly, given the young age of the participants, it is highly unlikely that these early abnormalities were the result of substantial accumulation of pathology (Aβ or tau). Tau pathology, in particular, first accumulates in the MTL [64,65] and this accumulation has been speculated to lead to abnormalities in MTL brain function [66]. If this aberration in brain function does occur before pathology, then changes in functional activity might be the driving force of further neurodegeneration and future impairments in cognition as argued by the Cascading Network Failure Hypothesis [67]. However, follow-up studies combining similar early functional abnormalities and measures of tau are needed to confirm whether early MTL abnormalities in brain function can occur in the absence of concomitant tau accumulation. 

The present study also aimed to test whether potential early abnormalities might be evident in temporal fluctuations of the BOLD signal as measured by standard deviation metrics (BOLD SD). Such measures of temporal fluctuations are novel and quickly growing to suggest that they might be used as a marker of brain health [41,42]. In addition to being a general marker of brain health, some researchers have proposed that early aberrations in hippocampal functioning might be due to an overactive or “noisy” system at rest [66,67]. To the extent that additional baseline noise might be captured by temporal variability, BOLD SD has potential to capture this dynamic function. While these novel markers show promise in other studies, they did not seem to be as sensitive as more traditional measures of mean activity in the present study. One possibility is that a difference in BOLD SD is a better marker of cognitive-aging processes than AD-related processes. Indeed, most research utilizing BOLD SD as an outcome measure has been in studies of normal cognitive aging [42,68]. Consistent with the notion of an age-specific sensitivity, a recent study found correlations between BOLD SD and age in a normal control sample, but not in a sample with a neurodevelopmental disorder [69].

We also found that family history of ADRD was associated with differences in brain structure. Studies investigating early genetic risk factors of AD in young adults have not consistently found differences in brain structure. Unlike those previous studies, we were able to detect abnormalities in hippocampal volume that were in the same region as the differences we found for brain function. Several factors may explain why we were able to find volume differences between these two groups. First, having a family history of ADRD sometimes reveals stronger effects than having the APOE ε4 allele, potentially making family history a more sensitive risk factor to detect alterations in brain structure than having the APOE ε4 allele. Second, our measure of hippocampal volume consisted of a ratio score that considered both the volumes of the inferior lateral ventricle and the hippocampus, called a hippocampal occupancy score. This score has been argued to be more sensitive to atrophy in a cross-sectional sample in relation to the progression of neurodegeneration [50]. The idea behind the creation of this score is that a fully intact hippocampus should occupy the majority of the space in the medial temporal lobe, with minimal enlargement of the surrounding ventricle. In contrast, as the hippocampus begins to deteriorate, the surrounding ventricle expands, thus occupying more space. Thus, this measure estimates the potential size of the hippocampus and, as in previous research, appears to be a sensitive indicator of early hippocampal decline [50]. Measures of white-matter integrity might also be a sensitive measure to detect such differences. Recently, structural connectivity analyses were shown to be reduced in middle-aged adults with a family history of ADRD in networks connected to the MTL [70].

In regard to the findings of hyperactivity, previous research investigating preclinical stages of AD and MCI has demonstrated hyperactivity in the hippocampus. However, a debate surrounds whether this hyperactivity is beneficial or detrimental to an individual. Some initial hypotheses proposed that the increased activity might serve a compensatory role in early stages of the disease [29,34]. Evidence for this idea comes, in part, from older adults with MCI that have smaller hippocampal volumes and greater hippocampal brain activation than older adults in similar stages, but no differences in episodic memory performance. In other words, greater brain activity may help ameliorate declines in memory performance that should be seen due to atrophy in the hippocampus. These findings in older adults resemble the findings here in younger adults: having a family history of ADRD is also associated with smaller hippocampal volumes and increases in brain activity in the hippocampus, but no differences in memory performance. However, these patterns do not necessitate that the processes are related to one another. Furthermore, compensatory arguments have been based on studies that use event-related fMRI and found such hyperactivity during successful memory trials. Because we found hyperactivity at rest rather than during successful task performance, no active compensatory processes would likely be occurring. 

In contrast, more recent studies have proposed that such hyperactivity may signal the beginning stages of hippocampal failure [29,67,71]. While many potential mechanisms at the neural level have been proposed to explain the deleterious view of hippocampal hyperactivity, one popular hypothesis points to the role of metabolic demands in specific brain regions such as the hippocampus. Specifically, some brain regions are highly connected to other brain regions, thus serving as “hubs.” Many of these regions overlap with the DMN. Repetitive or sustained use of these hubs requires extra metabolic demands, which might then lead to relatively more wear and tear than non-hub regions. It has been argued that this overuse might make these regions more vulnerable to downstream cellular and molecular consequences related to AD, including Aβ deposition [27,67]. Alternative models of detrimental hyperactivity have been proposed including an excitation/inhibition imbalance [36,37,38]. Such models propose that accumulation of Aβ deposition is the progenitor of these effects. Given that young adults likely have not accumulated Aβ deposition yet, the present findings argue against these models.

These hypotheses are difficult to disentangle because elevations of the BOLD signal are potentially confounded by age or disease-related changes that are non-neural in origin. For example, both age and disease can impact the coupling between neural activity and the vascular system—which is what the BOLD signal capitalizes on [72]. Alternatively, changes in resting hypoperfusion and metabolism might lead to an increased BOLD signal [73]. Thus, hyperactivity found in older adults might not be a sign of compensation or abnormal hippocampal functioning at all. However, by studying young adults, these confounds can be minimized. The findings in the present study provide evidence that hyperactivity can be found in relation to potential pathological processes (i.e., the risk conferred by having a parent with ADRD) when likely not confounded by non-neural effects. In light of this rationale, we believe our findings support the deleterious role of hyperactivity.

Despite the new insights that this study provides, it does have several limitations. As mentioned above, the primary limitation is the small sample size. Out of over a thousand recruited participants within the age range of 22–35, only 14 had a parent with AD. While the sample size is quite small, this population is also very unique. Not many young adults already have a parent with ADRD. To minimize type I errors due to small sample sizes, we took an a priori ROI approach targeting only one brain region theorized to be involved in the earliest stages of AD. In addition, we created a control group using propensity score matching, which has shown to be effective in reducing confounds between the comparison groups of interest and is especially suitable for matching uneven sample sizes [59]. Another limitation of the small sample size is the inability to differentiate between maternal and paternal family history of ADRD. Lastly, participants self-reported having a parent with ADRD and were not required to provide a confirmed medical diagnosis. Thus, future studies of this kind would benefit from verified medical diagnosis. 

## 5. Conclusions

The present study suggests that even young adults with a parent with ADRD show signs of abnormal hippocampal brain functioning and smaller hippocampal volumes. These patterns resemble those of older adults in early stages of AD, suggesting a very high risk for developing ADRD in the future and potential accelerated brain aging [74]. This study provides evidence that fMRI during rest has the potential to serve as an early biomarker of disease progression and can be used to detect at-risk individuals for late-life dementia. Moreover, the present findings suggest that lifestyle interventions should be implemented in people with a family history of ADRD much earlier in life than previously believed. However, unlike the unique population tested here, most individuals will not know whether they are at risk until middle age when their parents are old enough to show differential signs of “normal aging” and “pathological aging” such as AD. Thus, by the time people are aware that they are at risk, their brains might already evidence aberrations in both brain function and brain size in one of the most well-known regions to decline early in AD—the hippocampus.

## Figures and Tables

**Figure 1 brainsci-12-00496-f001:**
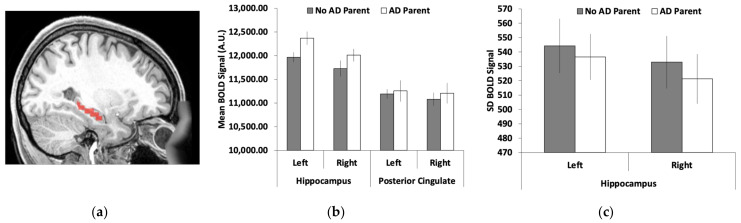
Functional magnetic resonance imaging (fMRI) measures of left and right hippocampi: (**a**) the hippocampal region of interest (ROI); (**b**) the mean blood-oxygen-level-dependent (BOLD) signal was greater for young adults who had a parent with Alzheimer’s disease/dementia (AD) than those without AD in the left hippocampus with the same trends, albeit non-significant, in the right hippocampus; (**c**) no significant effects for the standard deviation of the BOLD signal between the two groups. Brain images represent regions of interest overlaid on a representative younger adult brain normalized to Montreal Neurological Institute (MNI) space. Error bars represent the standard error of the mean (SEM).

**Figure 2 brainsci-12-00496-f002:**
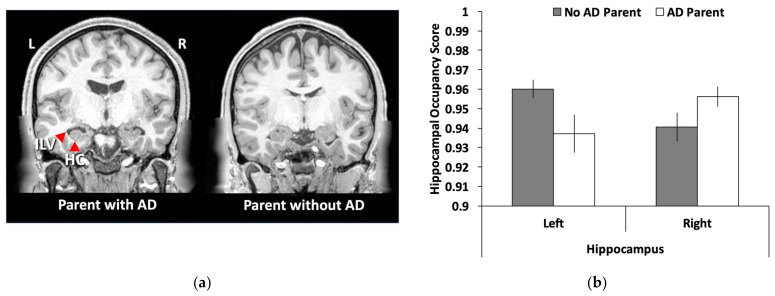
Hippocampal volume differences in young adults with and without a parent with ADRD: (**a**) a representative sample of hippocampal volume differences between the two groups with the left anatomical brain image showing a young adult with a parent with AD that has an enlarged inferior lateral ventricle and smaller hippocampus than the young adult on the right; (**b**) a bar graph showing that young adults with a parent with Alzheimer’s disease/dementia (AD) had smaller hippocampal occupancy scores [hippocampal volume/(hippocampal volume + inferior lateral ventricle)] than young adults with a parent without AD. No differences were found for the right hippocampus. Error bars represent the standard error of the mean (SEM). ILV = inferior lateral ventricle; HC = hippocampus; L = left; R = right.

**Table 1 brainsci-12-00496-t001:** Sample Characteristics.

	Parent without AD	Parent with AD
N	14	14
Age (years)	28.93 (4.34)	29.86 (4.62)
Age Range	22–35	22–35
Known APOE E4 Positive (N)	2	4
Sex (M/F)	6/8	7/7
Race (White)	79%	79%
Ethnicity (Hispanic)	7%	7%
Education (years)	13.79 (1.67)	13.86 (1.96)
Education Range	11–17	11–17
Employment Status		
Not Working	14%	36%
Part Time	43%	14%
Full Time	43%	50%
Parents with Depression	71%	71%
Parents with Bipolar Disorder	36%	29%
Mini_Mental State Exam Score	29.29 (0.91)	29.00 (0.96)
Mini_Mental State Exam Range	27–30	27–30
Reading Ability Score	116.04 (7.31)	116.50 (11.62)
Reading Ability Range	103.51–141.32	100.63–141.32
Verbal Memory Score	35.21 (2.91)	34.92 (3.34)
Verbal Memory Range	30–39	30–39
Visual Memory Score	108.60 (9.70)	107.07 (10.99)
Visual Memory Range	88.97–121.82	87.10–125.71

## Data Availability

Young adult data from the Human Connectome Project are publicly available at https://www.humanconnectome.org/study/hcp-young-adult/data-releases/ accessed on 01 March 2017.

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
