# Peer review of "Young Adults with a Parent with Dementia Show Early Abnormalities in Brain Activity and Brain Volume in the Hippocampus: A Matched Case-Control Study"

_brainsci, 2022, doi:10.3390/brainsci12040496_

Round 1

Reviewer 1 Report

This manuscript requires some essential editing. Here are some editing that needs to be considered:

  1. Authors should add some recent references to support the hypothesis and discussion of their study.
  2. Explain the abbreviations, at the first time they are used in the manuscript. For example, MTL, MCI etc.
  3. There is no description how the authors measured the hippocampus volume and calculated the occupancy score?
  4. In results section, Subheading states Brain volume, though the description is only about the hippocampus volume. 

Author Response

We thank the reviewer for their comments. Below are out responses.

1. Authors should add some recent references to support the hypothesis and discussion of their study.

We have added newer references to the Introduction and Discussion. However, upon reconducting a literature search, we did not find any new studies investigating a family history of AD in young adults, highlighting the uniqueness of the current study.

2. Explain the abbreviations, at the first time they are used in the manuscript. For example, MTL, MCI etc.

We have gone through the manuscript and added abbreviations when first mentioned.

3. There is no description how the authors measured the hippocampus volume and calculated the occupancy score?

On p. 4, we describe that we used Freesurfer to calculate volumes for the hippocampus and calculated the occupancy score: “To assess hippocampal volume, a hippocampal occupancy score was created [50] by taking the values for the left and right hippocampal volumes and dividing them by the sum of the hippocampal volume and the inferior lateral ventricle volume for each hemisphere, separately. The left and right hippocampal occupancy scores were then corrected using intracranial volume estimates from Freesurfer.”

4. In results section, Subheading states Brain volume, though the description is only about the hippocampus volume. 

We have now made the subheaders more specific.

Reviewer 2 Report

The manuscript presents the results of an interesting work on a very common pathology today.

The setting of the scientific methodology is correct and the results support the assumptions.

It would be interesting to better specify the technical parameters of the BOLD sequences used for the study.

Author Response

We thank the reviewer for their comments. Below are out responses.

1. The manuscript presents the results of an interesting work on a very common pathology today.

Thank you.

2. The setting of the scientific methodology is correct and the results support the assumptions.

Thank you.

3. It would be interesting to better specify the technical parameters of the BOLD sequences used for the study.

On p. 4, We state the following: “BOLD fMRI data were acquired using a T2*-weighted gradient-echo EPI sequence with 72 axial slices per volume, 104 x 90 matrix (2.0×2.0×2.0 mm3), FOV=208 mm, TE=33.1 ms, TR=720 ms, FA=52°. Across four scanning sessions of 15 minutes each, a total of 4800 frames were acquired. Participants were instructed to keep their eyes open and focused on a bright cross hair on a dark background. Across sessions, oblique axial acquisitions alternated between phase encoding in a right-to-left direction (two runs) and phase encoding in a left-to-right direction (two runs).”